# Failure Mechanism of Anti-Dip Layered Soft Rock Slope under Rainfall and Excavation Conditions

Jun Jia [1,2], Xiangjun Pei [2], Gang Liu [1,\*], Guojun Cai [2], Xiaopeng Guo [1] and Bo Hong [1]

[1]   Key Laboratory for Geo-Hazard in Loess Area, Ministry of Natural Resources, Xi'an Center of China Geological Survey, Xi'an 710119, China; jiajun@mail.cgs.gov.cn (J.J.); sjcgxp@163.com (X.G.); hongbo@chd.edu.cn (B.H.)
[2]   State Key Laboratory of Geohazard Prevention and Geoenvironment Protection, Chengdu University of Technology, Chengdu 610059, China; peixj0119@tom.com (X.P.); caiguojun@cdut.cn (G.C.)
\*   Correspondence: liugang_iggcas@163.com

**Abstract:** The phenomenon of toppling deformation and failure is common in slopes with anti-dip structures, especially in soft metamorphic rock slopes. This paper aims to explore the instability mechanism of anti-dip layered soft metamorphic rock landslides. Taking the slope of a mining area in the southern Qinling Mountains of China as a geological prototype, a large-scale centrifuge model test and a numerical simulation based on the combined finite and discrete element method (FDEM) were performed. The deformation and failure process, failure mode, and failure path of the slope under rainfall and excavation conditions were simulated. The results show that both the physical centrifuge model test and the new numerical model test can simulate the instability process of anti-dip layered soft metamorphic rock slopes, and the phenomena simulated by the two methods are also very close. Rainfall mainly weakens the mechanical properties of rock, while the excavation at the slope toe mainly changes the stress field distribution and provides space for slope deformation, both of which accelerate the instability of the anti-dip soft metamorphic rock slope. The failure process of an anti-dip layered soft rock slope can be described as follows: bending of the rock layer–tensile fracture along the layer–flexural toppling and cracking perpendicular to the rock layer–extension and penetration of the tensile fracture surface–sliding and instability of the slope.

**Keywords:** anti-dip; layered soft metamorphic rock; centrifuge model test; FDEM; slope instability; rainfall and excavation

## 1. Introduction

The unique regional tectonic setting has created rich mineral resources in the Qinling region [1]. In recent years, with the development of the economy, the demand for mineral resources has been increasing, and thus mining activities in the southern Qinling Mountains are becoming increasingly intense. However, mining engineering activities also promote the development of geological disasters [2,3]. At the same time, the special geographical environment and climatic factors have created unique rainfall characteristics in the southern Qinling Mountains, with an average annual rainfall of 600–1300 mm. Rainfall has also become a major factor in the frequent occurrence of geological disasters in the region [4,5]. The geological disasters in the southern Qinling Mountains have caused enormous threats and losses to the safety of human life and property, and the forms of disaster prevention and mitigation in this region are severe.

A large amount of soft metamorphic rock has developed in the southern mountain area of the Qinling Mountains, which is prone to softening and causing slope instability under engineering disturbance and rainfall infiltration. Previous studies have focused on the stability of bedding slopes, but through geological surveys, it has been found that the deformation and failure of inverted slopes are also very common. Therefore, conducting

research on the instability mechanism of anti-dip soft metamorphic rock landslides in mining areas under rainfall conditions can provide a theoretical basis for formulating disaster prevention measures and has important theoretical and practical significance.

Physical model testing is a commonly used method for studying slope deformation and failure [6–26]. Through model tests, it is possible to directly observe and record the deformation and failure phenomena of rock masses and display the entire dynamic process of slope instability, which is very effective for understanding the mechanism of slope instability. For example, Zheng et al. [18] used a large centrifuge model experiment to study the influence of the slope angle on the process and mode of toppling deformation and failure; Bai et al. [13] studied the catastrophic process and genetic mechanism of red bed soft rock landslides in Wumeng Mountains through centrifuge model tests under rainfall conditions; Zhao et al. [11] explored the toppling deformation mechanism of the right bank inverted slope of the Tibet Zhala Hydropower Station through physical model tests and discussed effective methods for monitoring and warning of an inverted slope; Lin et al. [12] used small-scale simple model tests to study the failure process and characteristics of inclined and anti-dip layered rock slopes under rainfall conditions and explored the impact of rainfall and groundwater on slope instability; Zheng et al. [8] studied the toppling deformation mode and collapse formation mechanism of such slopes through bottom friction physical model tests, and considered the impact of underground mining; Yang et al. [7] disscussed the dynamic response characteristics and failure process of anti-dip slopes under earthquake action through shaking table tests; and Adhikary et al. [6] compared and studied the toppling deformation and failure modes of slope models with different materials. In summary, physical model experiments have achieved good results in simulating the process of slope instability. However, due to the complexity of the model's construction process, strict material selection, high difficulty in monitoring displacement and stress, long testing cycles, and high costs, this method is difficult to carry out on a large scale.

Numerical simulation is another commonly used method for studying slope deformation and failure. This method can record the changes in stress and displacement fields within the slope at any moment in real time and simulate the entire process of slope deformation and instability. Several numerical simulation methods have been successfully applied to the study of anti-dip slopes [27–37]. As early as the 1990s, Orr et al. [27] used the finite element analysis method to carry out an instability analysis of the toppling failure of open pit mines, and Coggan et al. [28] studied the stability of anti-dip slate stope slopes using the discrete element method. Zhang et al. [31] used the particle discrete element method to study the influence of structural plane parameters on the deformation and failure mechanism of anti-dip slopes and pointed out that deformation usually occurs first at the top of the slope, and failure usually begins at the foot of the slope. Ma et al. [32] used the discrete element method to simulate and analyze the "S" type deformation mechanism of the slope under the softening action of reservoir water. Current numerical analysis is still dominated by the finite element method, finite difference method, traditional discrete element method, and particle discrete element method. There are problems, such as considering the limited number of structural surfaces and the inability to achieve crack propagation. Although the numerical simulation method is a repeatable and economical analysis method, the current research on anti-dip slope simulation is mainly based on finite element, finite difference, and traditional discrete element methods [31,38–40]. Some scholars have also used new numerical methods, such as DDA, CDEM, and FDEM, which can effectively describe the fracture of rock or soil to conduct relevant research [41–45]. Liu et al. [44] demonstrated the effectiveness of the FDEM method throughout the entire process simulation of anti-dip slope instability. Song et al. [45] adopted the CDEM method to conduct seismic damage and dynamic analysis on a high, steep, anti-dip rock slope. However, the application of the new methods above in analyzing the instability process and mechanism of the anti-dip slope is still relatively weak. As an essential supplement to

physical model experiments, the new numerical methods could complement each other in studying the mechanism of slope instability.

In order to study the failure mechanism of an anti-dip layered soft rock slope under the combined action of rainfall and excavation, based on the geological prototype, the research method of combining physical model test and numerical simulation was adopted to obtain the deformation and failure process of the slope, as well as the stress and displacement variation curves at monitoring points. By comparing the results of the physical model test and numerical model test, the failure process and mode of anti-dip layered soft metamorphic rock slope are summarized. The research results are helpful for further understanding the failure mechanism of such slopes in southern Qinling Mountain. Additionally, the deformation and failure process can guide the engineering treatment of anti-dip slopes at road slope cutting or building slope cutting.

## 2. Methods

### 2.1. Centrifuge Model Test

(1)   Test set-up

The principle of the centrifuge model test is to use a small proportion model to simulate the gravity stress of rock and soil mass via centrifugal force so that the model can achieve the same stress level and similar deformation as the prototype in the gravity field. The centrifugal force of $n$ times gravity should be applied to the model in the centrifugal field to simulate the self-gravity stress field after scaling $1/n$ of the model so as to simulate the deformation and failure characteristics of the prototype slope. The physical centrifuge model test was carried out using TLJ-500 geotechnical centrifuge of the State Key Laboratory of Geological Disaster Prevention and Geological Environmental Protection, Chengdu University of Technology, which is currently the largest capacity geotechnical centrifuge in China (Figure 1). It is composed of the host and the main console, the full digital DC speed regulating device, the driving system, the data acquisition and transmission system, the data processing system, the high-speed camera function, the video monitoring system, and the model box. The size of the model box is 100 cm long, 60 cm wide, and 70 cm high. The principal technical indicators are shown in Table 1.

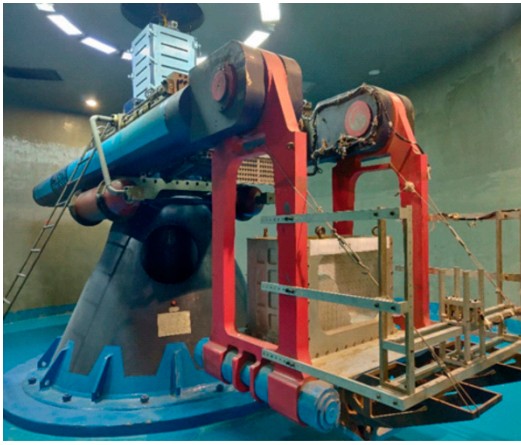

**Figure 1.** TLJ-500 geocentrifuge.

**Table 1.** Main technical indexes of TLJ-500 geocentrifuge.

| Effective Capacity (g·t) | GAL (g) | Effective Radius (m) | Size of Model Box (Length/Width/Height, cm) |
|---|---|---|---|
| 500 | 250 | 4.5 | 100/60/100 |

(2)    Test materials

The anti-dip stratified soft rock slope called Caiziba landslide in a mining area in Lueyang County, Shaanxi Province, was taken as the geological prototype (Figure 2). The slope inclination is 7°, and the dip is 45°; the trailing edge, the leading edge, and the height are 15 m, 21 m, and 31 m, respectively. The strata are the thin layered schist of Paleozoic Silurian superposition c member ($S_1d^c$), and the lithology is mainly medium-strong weathering phyllite with the occurrence of 350°∠70°. It is gray to gray-black and has fine textures. Small folds and some micro-cracks can be seen locally.

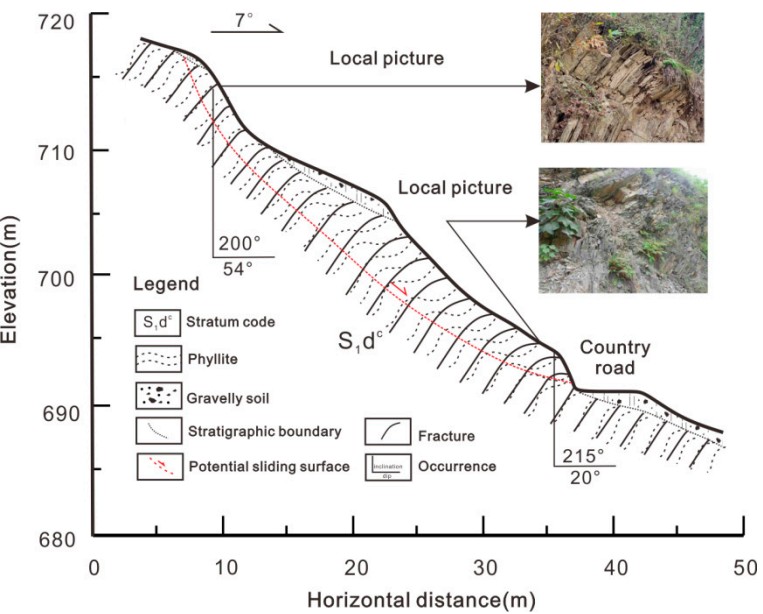

**Figure 2.** Photos of Caiziba landslide and its geological engineering profile.

According to the requirements of standard for test methods of engineering rock mass (GB/T 50266-2013) [46], the uniaxial compression test was carried out on the core sample with the size of φ50 × 100 mm, and the uniaxial compressive strength was 9.54~12.50 MPa. Powder diffraction analysis of the rock sample of Caiziba showed that its mineral composition is composed of mica, chlorite, quartz, and plagioclase, with chlorite and quartz as the main components (Table 2).

**Table 2.** Results of X-ray overall mineralogical analysis.

| Mineral Types | Mica | Kaolinite | Chlorite | Quartz | Feldspar | Plagioclase | Calcite | Dolomite | Gypsum | Pyrite | Borax |
|---|---|---|---|---|---|---|---|---|---|---|---|
| Mass Content (%) | 12.95 | ND | 47.31 | 35.43 | ND | 4.31 | ND | ND | ND | ND | ND |

Note: "ND" represents not detected.

Determining the similarity of physical model experiments is a prerequisite and foundation for obtaining reliable results and truly reflecting the physical and mechanical properties of the prototype [17]. Considering the complexity of the prototype slope, the slope characteristics of the anti-dip soft metamorphic rock are generalized. According to the similarity principle, combined with prototype ramp size, model box size, centrifuge capacity, and maximum centrifugal acceleration, geometric similarity ratio of $C_l$ = 1/120 (model/prototype) and centrifugal acceleration of 120 g were selected. Density, compressive strength, elastic modulus, internal friction angle, and cohesion were selected as similar parameters of the model control in the model test. First of all, the scale factor of length ($C_l$ = 1/120, model to prototype) was first determined based on the geological prototypes of slope in field and laboratory model box dimensions, and hence the $C_u = C_l$ = 1/120. That is to say, the centrifuge acceleration is 120 g (i.e., $C_a$ = 120/1) based on the similarity principle of

centrifuge. To accurately model the property of rock layer, the density of similar material is selected as the same as the rock in the field, namely, $C_\rho = 1/1$. Then, the scale factor of cohesion is calculated as $C_\sigma = C_\rho C_l C_a = 1/1$. As the unit of the modulus is identical to that of the cohesion, the $C_E = C_\sigma = 1/1$. The friction angle, strain, and Poisson ratio are dimensionless parameters, and thus $C_\varphi = C_\epsilon = C_\mu = 1/1$. The similarity relationship used in the centrifuge model test is listed in Table 3.

**Table 3.** Scale factors used in centrifuge model tests.

| Physical Quantity | Symbol | Similarity Ratio Symbol | Scale Factor (Model/Prototype) |
|---|---|---|---|
| Length | $l$ | $C_l$ | 1/120 |
| Density | $\rho$ | $C_\rho$ | 1/1 |
| Elastic modulus | $E$ | $C_E$ | 1/1 |
| Acceleration | $a$ | $C_a$ | 120/1 |
| Displacement | $u$ | $C_u$ | 1/120 |
| Cohesion | $c$ | $C_c$ | 1/1 |
| Internal friction angle | $\varphi$ | $C_\varphi$ | 1/1 |
| Stress | $\sigma$ | $C_\sigma$ | 1/1 |
| Strain | $\varepsilon$ | $C_\varepsilon$ | 1/1 |
| Poisson ratio | $\mu$ | $C_\mu$ | 1/1 |

In order to simulate real landslide deformation failure, we based our experiment on the physical and mechanical properties and mineral composition of phyllite in Rapeseed bar, combined with the comparative selection experience of similar materials in relevant literature [47–49]. Quartz sand, gypsum, cement, and water were selected as similar materials to soft metamorphic rocks (Table 4). The optimal ratio of similar materials was determined by orthogonal uniaxial compression test, and the ratio was determined as quartz sand:gypsum:cement = 16:16:1.

**Table 4.** Main physico-mechanical parameters of prototypes and similar materials.

| Type | Material Lithology | Density $\rho$/(kg/m$^3$) | Elastic Modulus $E$/MPa | Compressive Strength $\sigma$/(MPa) | Cohesion $c$/(kPa) | Internal Friction Angle $\varphi$/(°) |
|---|---|---|---|---|---|---|
| Prototype | Phyllite | 2500 | 2500 | 9.54~12.50 | | |
| | Interlayer bonding | - | - | - | 46.52 | 2.63 |
| Model | Phyllite | 2580 | 2480 | 12.53 | | |
| | Interlayer bonding | - | - | - | 46.15 | 2.34 |

(3)　Model construction and testing procedure

Based on the geometric characteristics of Caiziba landslide, the generalized model slope is a straight slope with a slope angle of 50° and a dip angle of 60°. The dimensions of the slope model in Figure 3 are determined based on a combination of scale factors, geological prototypes of slope in field, and laboratory model box dimensions. According to the similarity ratio and the size of test box, the design size of the anti-dip rock slope model is determined as 61 cm in length, 50 cm in width, and 48 cm in height.

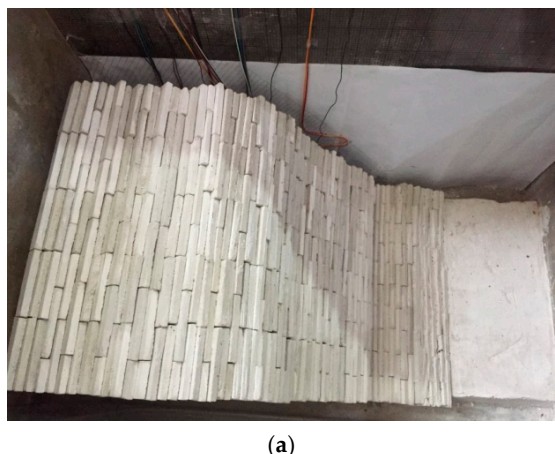 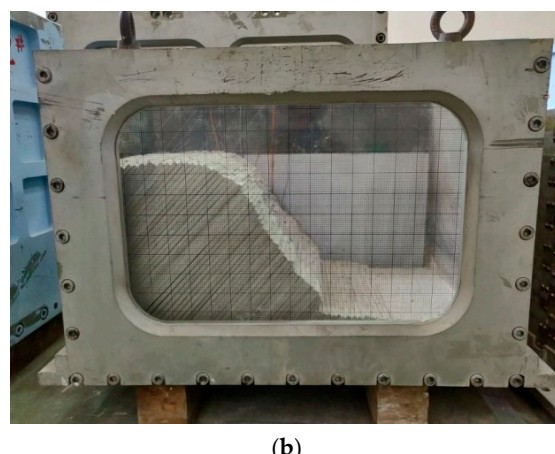

(**a**) (**b**)

**Figure 3.** Model construction: (**a**) top view of slope model; (**b**) completed model in the test box.

The centrifuge test model is established based on the topographic lines of the slope prototype. According to the size requirements of different locations of the model, the prefabricated test blocks with a thickness of 10 cm were cut and stacked. The test blocks were filled with bonding similar materials. Similar bonding materials were arranged in proportion with quartz sand, gypsum, cement, and water. The bonding force between the test blocks was simulated by filling similar bonding materials [18]. After the model was stacked, it was covered with film and maintained for 3 days. The completed stacked model is shown in Figure 3. Similar materials were filled between test blocks to simulate the bonding state between rock layers. It should be mentioned that in practice, there is a thin layer of soil on the surface of the rock layer, but it has almost no impact on the stability of the entire slope. The physical and mechanical properties of the anti-dip layered rock control the stability of the slope. Therefore, the slope was generalized in the model test without considering the superficial soil. During the model construction process, 4 soil pressure sensors (T1 to T4) and 10 strain gauges (Y1 to Y10) were installed. The layout of the sensors and strain gauges is shown in Figure 4. At the same time, a high-resolution PIV high-speed camera was installed directly above the model box. The entire deformation and failure processes of slope in the test were recorded. Black grid paper was pasted on the glass plate of the model box. Through image measurement technology, relative displacement vector data can be obtained by comparing the spatial coordinates of various points on the slope before and after the test.

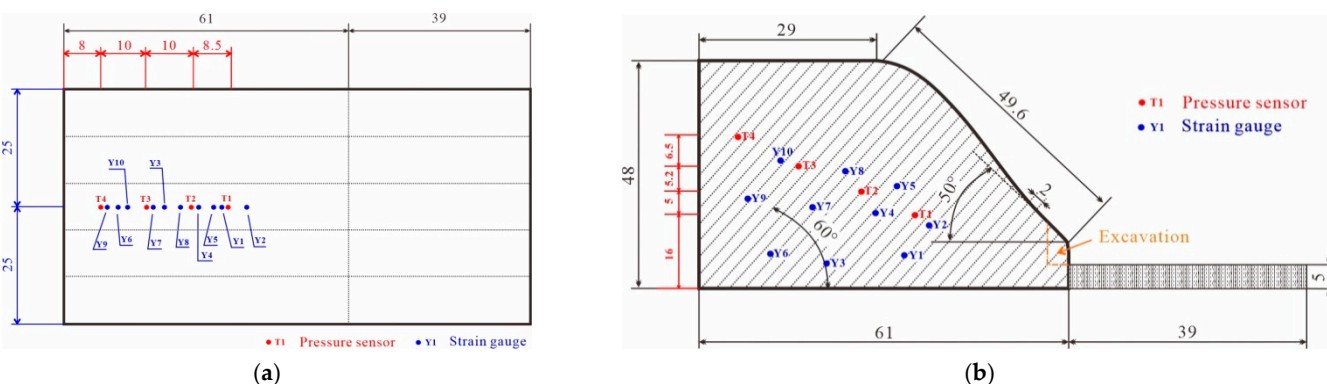

(**a**) (**b**)

**Figure 4.** Sensor Layout: (**a**) Top view; (**b**) Side view.

Mian–Lue–Ning area is located in the subtropical monsoon climate zone, and rainfall is an important factor inducing landslide disasters in this area. Therefore, the impact of rainfall conditions should be considered in the model test. Considering that the rainfall is uneven during the centrifuge operation, the rainfall intensity and time are difficult to

control, and the rainfall is difficult to fully infiltrate. The controllable pneumatic spray can be used to spray water mist on the surface of the model slope to simulate the rainfall process so that the rainfall can penetrate evenly and gradually increase the saturation of the slope. With consideration of the local annual rainfall value and similarity ratio, it was finally determined that the amount of water sprayed is $1.572 \times 10^6$ mm$^3$. The loading process was carried out by centrifuging step by step, and the next stage of loading was carried out after each stage was stable for 5 min. The entire loading stage was divided into four stages, as shown in Figure 5.

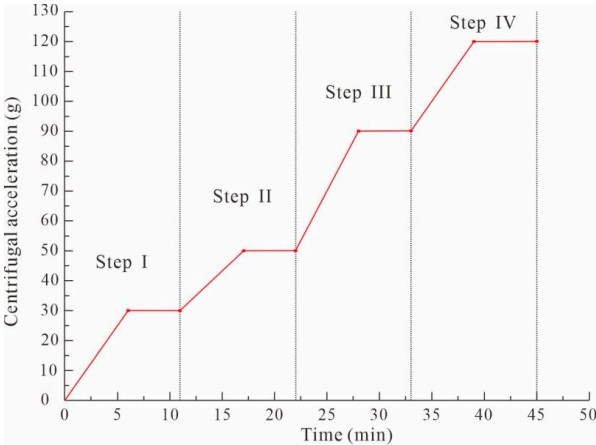

**Figure 5.** Centrifuge loading scheme.

### 2.2. Numerical Model Test

Based on the prototype of the Caiziba anti-dip layered soft metamorphic rock slope in Lueyang County and the centrifuge model, a numerical model was established to study the deformation and failure process of the anti-dip layered soft rock slope. The study was conducted using a Y-Mat program based on a hybrid finite-discrete element method (FDEM).

(1)     Fundamentals of Y-Mat program

The theoretical basis of the Y-Mat program is the hybrid finite-discrete element method first proposed by Munjiza et al. [50], which can effectively solve material deformation and fracture problems. The fundamentals of the method can be described as follows. First, the model is divided into finite element meshes. Second, adhesive crack model elements are embedded between any two adjacent mesh elements, as shown in Figure 6. Finally, the finite element algorithm based on the continuous medium theory can be used inside each element to describe the deformation of the model, and a discrete element algorithm based on the discontinuous medium theory can be used between elements to describe the formation and propagation of cracks [44]. The equation of motion is expressed as

$$M\ddot{x} + C\dot{x} - f_{int}(x) - f_{ext}(x) - f_c(x) = 0 \tag{1}$$

where M is the mass matrix; C is a viscous damping matrix; $\ddot{x}$, $\dot{x}$, and x are the acceleration, velocity, and displacement of the node, respectively; $f_{int}$ is the internal force, including the elastic deformation force and the cohesive force of the joint element; $f_{ext}$ is the applied external force; and $f_c$ is the contact force of each element.

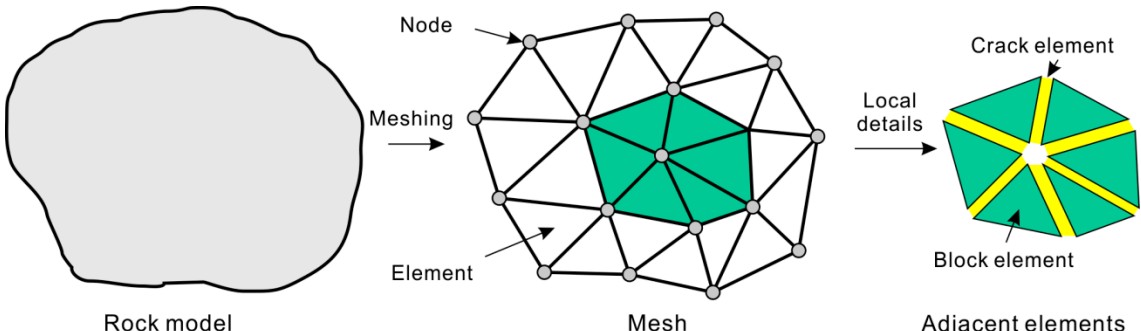

**Figure 6.** Mesh generation in FDEM.

Y-Mat program is based on the original program Y code. Additionally, it has been improved in three aspects, including a simpler and more efficient algorithm to calculate the contact force, an algorithm for tangential contact force closer to the actual physical process, and a plastic yielding criterion (e.g., Mohr-Coulomb) to modify the elastic stress for fitting the mechanical behavior of elastoplastic materials. Then, a program integrating the core algorithm, pre-processing, and post-processing was written on the MATLAB platform to facilitate numerical simulation.

(2)　Numerical model and scheme

Based on the actual shape of the Caiziba landslide and the geometric characteristics of the physical model test, a numerical model was established, as shown in Figure 7. The length, the left height, and the right height are 60 m, 38 m, and 8 m, respectively. The boundary condition of this model is to fix the normal displacement of the left and right sides and the bottom. Five monitoring points are set inside the slope to record deformation and stress changes.

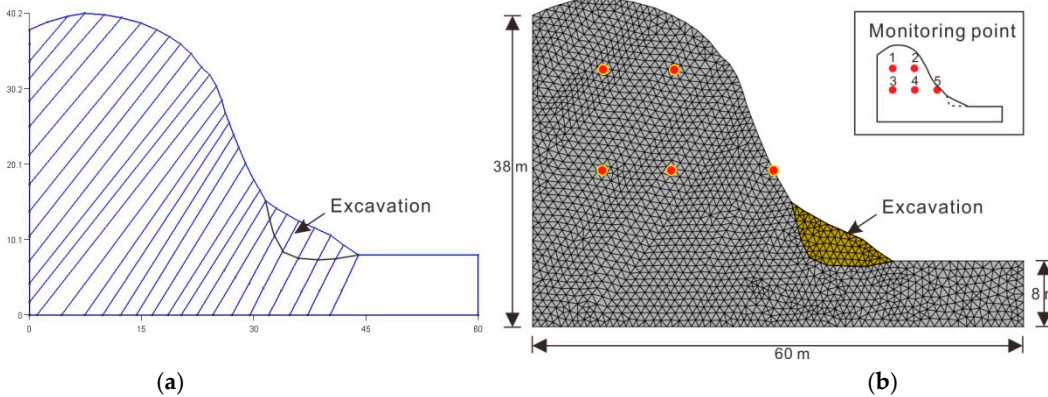

**Figure 7.** Numerical model of anti-dip layered slope: (**a**) Model geometry; (**b**) Mesh and monitoring points.

Based on the laboratory tests conducted by the authors and the results of relevant mechanical tests in the area, the numerical calculation parameters were determined through trial calculation, as shown in Table 5. The strength reduction method was employed to simulate the effect of rainfall.

**Table 5.** Numerical model test parameters.

| Element Category | Parameter Type | Layer | | Interlayer | |
|---|---|---|---|---|---|
| | | Non-Rainfall | Rainfall | Non-Rainfall | Rainfall |
| Elements | Young's modulus (GPa) | 2.5 | 2.0 | — | — |
| | Poisson's ratio (-) | 0.25 | 0.27 | — | — |
| | Density (kg/m$^3$) | 2500 | 2650 | — | — |
| | Cohesion (MPa) | 1.18 | 1.0 | — | — |
| | Friction angle (°) | 46 | 35 | — | — |
| | Tensile strength (MPa) | 1.4 | 1.1 | — | — |
| Bonding crack elements | Cohesion (MPa) | 1.18 | 1.0 | 0.5 | 0.3 |
| | Maximum static friction angle (°) | 50 | 42 | 40 | 30 |
| | Sliding friction angle (°) | 46 | 35 | 30 | 20 |
| | Residual friction angle (°) | 35 | 28 | 20 | 10 |
| | Tensile strength (MPa) | 1.4 | 1.1 | 0.84 | 0.5 |
| | Mode I fracture energy release rate (N/m) | $2.3 \times 10^4$ | $2.1 \times 10^4$ | $1.4 \times 10^3$ | $1.1 \times 10^3$ |
| | Mode II fracture energy release rate (N/m) | $1.2 \times 10^5$ | $1.05 \times 10^5$ | $1.4 \times 10^4$ | $1.1 \times 10^4$ |
| | Fracture penalty (Pa) | $2.8 \times 10^9$ | $2.3 \times 10^9$ | $1.68 \times 10^9$ | $1.35 \times 10^9$ |
| | Normal contact penalty (Pa) | $2.8 \times 10^9$ | $2.3 \times 10^9$ | $1.68 \times 10^9$ | $1.35 \times 10^9$ |

First of all, the Y-Mat program was used to calculate the self-weight balance of the model. Second, a portion of the soft rock at the toe of the slope was excavated to represent the actual excavation (Figure 7). Then, rainfall was applied to the model by reducing the mechanical parameters of soft rock. A mechanical calculation was conducted to obtain the deformation and instability process of the slope. Finally, the mechanical calculation was performed again to calculate the deformation and instability process.

## 3. Results

### 3.1. Centrifuge Model Test

(1) Deformation and failure process of the slope

Based on the field review survey and field reconnaissance, a physical model test was conducted by generalizing the typical failure mode of landslide in the field. The photos of the initial model are shown in Figure 8a. It should be noted that the rock mass within a certain range of the model slope toe was cut to simulate the actual conditions of erosion or excavation at the slope toe.

By using the high-speed camera, the four most representative images of the slope model were selected from the captured photos, as shown in Figure 8b–e. It can be seen from the figure that at the initial stage of the loading (0–30 g), the bedding rocks began to bend as a whole under the action of gravity. This provides the space for the deformation at the rear of the slope, leading to significant settlement deformation at the rear edge of the slope and forming settlement grooves (see Figure 8b). At the same time, accompanied by interlayer shear dislocation, a small number of tensile cracks were locally generated, but there was no sign of shear fracture. With the gradual increase in centrifugal acceleration (30–50 g), apart from shear action, the tensile effect between rock layers gradually became strong, and multiple obvious tensile cracks appeared in the middle part of the slope. At the toe of the slope, the rock stratum underwent stress concentration under the action of self-weight and strong compression of the overlying rock stratum, leading to the flexural toppling and the first local collapse of the rock mass. The overlying rock layer lost some support at the toe of the slope, and under the action of gravity, the toppling deformation intensified, causing bending-toppling. When the acceleration increased from 50 g to 90 g, the slope toppling further developed, the bending degree of the rock stratum gradually increased, and the interlayer tension phenomenon became more obvious. Intermittent tension cracks were generated inside the rock mass, which toppled towards the free surface, resulting in a wider range of fractures, and meanwhile, the rock mass underwent a second collapse. Finally, after the centrifugal acceleration reached 120 g, the settlement and bending deformation of the rock stratum further intensified, and a third and wider range of rock stratum fracture failure occurred in the upper part of the slope.

Based on the analyses of the deformation and failure process of the anti-dip layered soft rock slope, it can be observed that the rock stratum fracture occurred at the slope toe first and gradually produced a gradual rock stratum fracture from the slope toe to the top of the slope. Finally, due to the spatial accumulation caused by the fracture of the lower rock stratum, the rock stratum on the top of the slope toppled and broke, forming relatively obvious tensile cracks. As the tensile crack extended, the sliding force along the sliding surface exceeded the shear resistance, and then the rock layers quickly slid down.

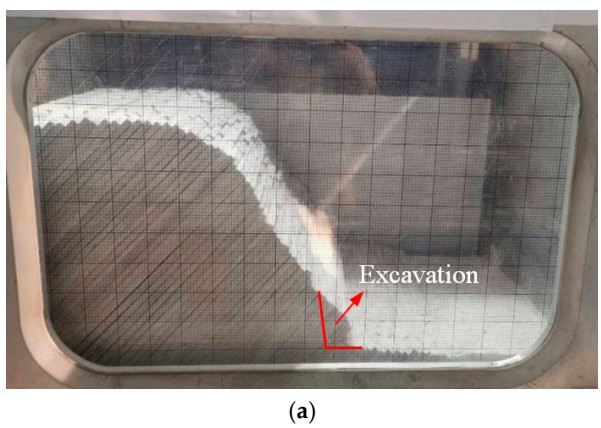

(a)

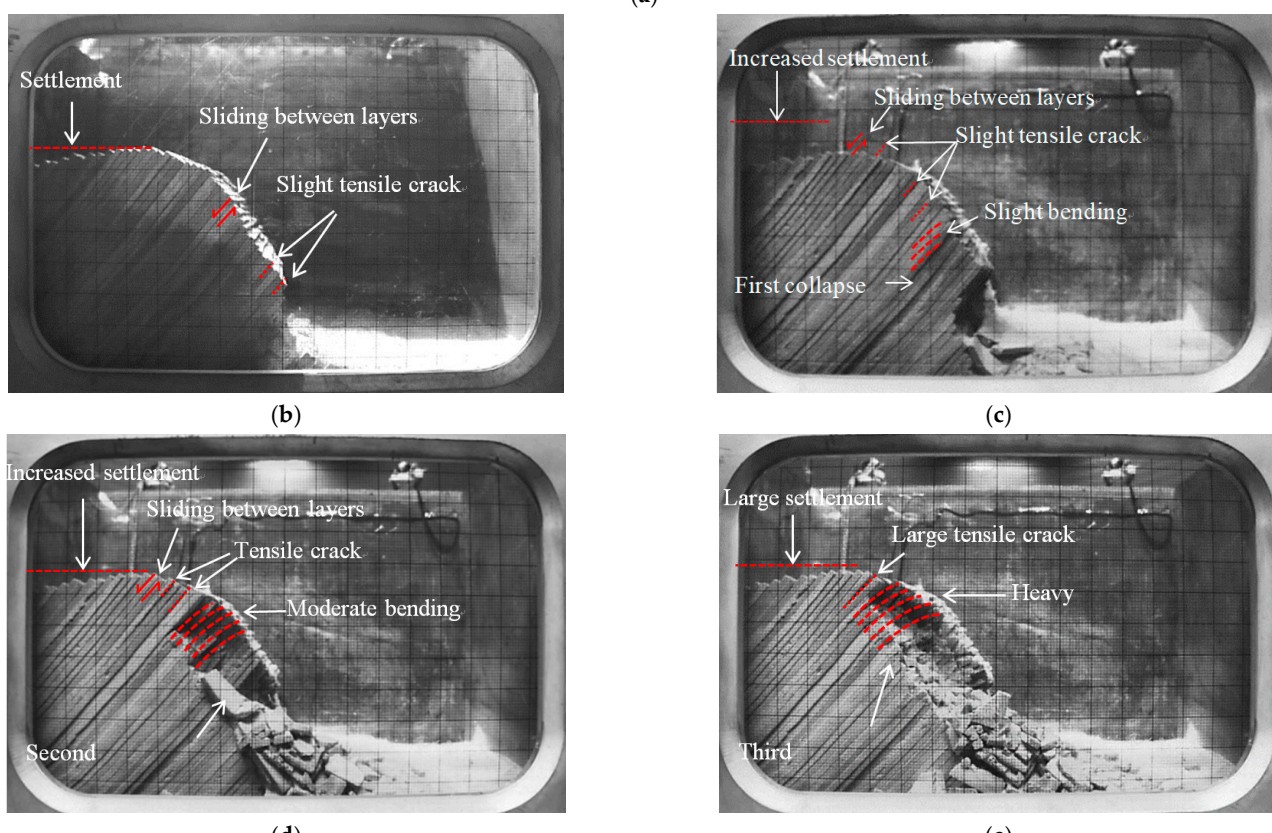

**Figure 8.** Deformation and failure process of anti-dip layered soft rock slope: (**a**) 0 g; (**b**) 30 g; (**c**) 50 g; (**d**) 90 g; (**e**) 120 g.

(2)  Monitoring results and analyses

●  Soil pressure

A total of four earth pressure gauges were used in this model test, with T1 and T2 at the toe of the front edge of the model and T3 and T4 inside the rear edge of the model. The variation in the earth's pressure with time is shown in Figure 9, and the specific layout

is presented in Figure 4. In general, the variation of soil pressure presents an increasing trend from the top to the toe of the slope. With the continuation of loading time, the centrifuge acceleration continuously increases, and the overall trend of soil pressure at each monitoring point increases, but the amplitude of soil pressure change is different. It reaches its maximum at about 90 g and then gradually decreases to 0 as the centrifuge acceleration decreases. Once the rock layers fracture and collapse, the earth's pressure decreases abruptly, namely, stress equal to zero.

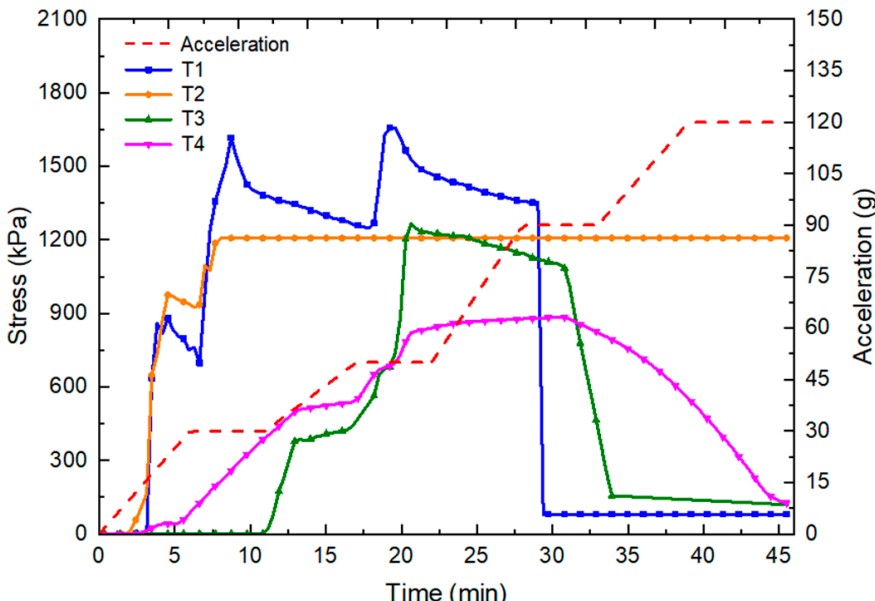

**Figure 9.** Variation curves of earth pressures with time at each monitoring point.

At a constant centrifugal acceleration of 10 g for 2.5 min, the T4 earth pressure basically remains unchanged and then increases rapidly. The increase in earth pressure indicates that settlement occurs at the top of the slope at this time. At 120 g, it reaches a peak value, and the maximum peak earth pressure is 910.3 kPa. As the loading time increases, it gradually decreases, indicating that tensile cracks have already occurred at the top of the slope.

Before reaching the centrifugal acceleration of 50 g, the T3 earth pressure remained unchanged at 0, indicating that no voids were compressed inside the slope during this period of time. Before the centrifugal acceleration of 90 g, the earth pressure rapidly increases, indicating that the internal joints, cracks, voids, etc., of the slope begin to compress, and the overall slope produces the compaction. When the centrifugal acceleration is at a constant speed of 90 g for 5 min, the earth pressure basically remains unchanged and then increases rapidly. At 120 g, the peak value is reached, and the maximum peak earth pressure is 1250 kPa. As the loading time increases, it gradually decreases, indicating that bending deformation occurred at the middle part of the slope at this time.

Before reaching the centrifugal acceleration of 20 g, the T2 earth's pressure rapidly increases, indicating that the shear force of the soil at the slope toe increases at this time. At a centrifugal acceleration of 20 g, the earth's pressure suddenly changes and rapidly decreases. It is believed that under the action of self-weight stress, local damage occurs within the slope, and energy is released accordingly, resulting in a rapid decrease in earth pressure. Afterward, the earth's pressure increases rapidly. At a peak value of 50 g, the maximum peak earth pressure inside the slope reaches 1203 kPa.

The variation trend of T1 is the same as T2 before centrifugal acceleration reaches 50 g. As the centrifugal acceleration increases from 50 g to 90 g, the earth's pressure decreases, indicating that under the action of self-weight stress, local damage occurs at the slope toe, resulting in the release of energy and a rapid decrease in earth pressure. At a centrifugal acceleration of 90 g to 120 g, the earth's pressure generally presents an increasing trend.

Reaching a peak value at 120 g, a maximum peak earth pressure is 1684 kPa. When the centrifuge acceleration is at a constant speed of 120 g, the T1 earth pressure drops to 1391 kPa, indicating that the slope has been damaged at this time and partial damage has occurred at the toe of the slope. Under the influence of the self-weight stress, a small range of sliding occurs, and bending tensile cracks occur.

- Strain

A total of 10 strain gauges were used in this model test, and 2 strain gauges (Y2, Y9) were damaged during the test. Therefore, no data were collected at Y2 and Y9. The strain change diagram for each position is shown in Figure 10, where Y1 is located at the front edge of the model; Y3, Y4, Y5, Y6, Y7, and Y8 are located in the middle of the model; and Y10 is located at the rear edge of the model. In Figure 10, the value of the left y coordinate is the strain at each monitoring point, and the value of the right y coordinate is the centrifuge acceleration. The loading process is carried out by centrifuging step by step, and the next stage of loading is performed after each stage is stable for 5 min. The whole loading stage is divided into four stages with accelerations of 30 g, 50 g, 90 g, and 120 g, respectively. Overall, the strain variation presents an increasing trend from the top to the toe of the slope. With the change in loading time, the centrifugal acceleration increases continuously, and the overall trend of strain change at each measuring point increases accordingly. It reaches its maximum at 120 g and then remains basically unchanged or slightly decreases as the centrifugal acceleration decreases.

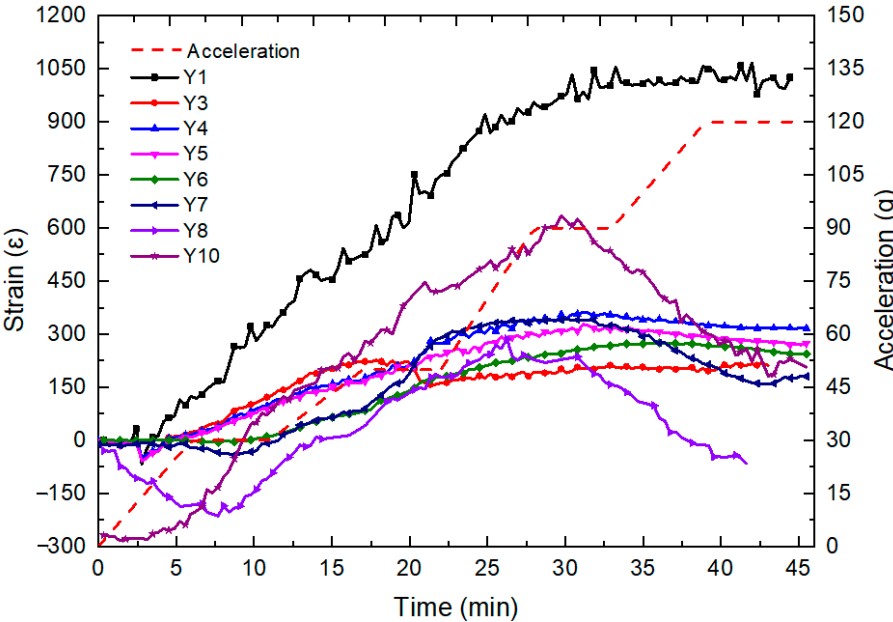

**Figure 10.** Variation curves of strains with time at each monitoring point.

Before reaching the centrifugal acceleration of 120 g, the Y10 strain value maintains an increasing trend, and the increase in strain indicates that settlement occurs at the top of the slope at this time. At 120 g, it reaches a peak value with a maximum peak strain of 620. As the loading time increases, it gradually decreases, indicating that tensile cracks have already occurred at the top of the slope.

The strain of Y3, Y4, Y5, Y6, Y7, and Y8 increases continuously before the centrifugal acceleration of 120 g, indicating that at this time, the internal rock strata of the slope experience sliding dislocation, and the overall slope produces compaction. At 120 g, the peak strain was reached, and the maximum peak strain was 211, 363, 354, 297, 345, and 288, respectively. As the loading time increases, it gradually decreases, indicating that bending deformation occurred in the middle part of the slope at this time.

Before the centrifugal acceleration of 120 g, the Y1 strain increased, indicating that the shear force of the soil at the slope toe increased at this time. The slope toe undergoes local damage under the action of self-weight stress, and the energy is released accordingly. The peak value is reached at 120 g, and the maximum peak strain inside the slope reaches 1050. When the Y1 strain gauge is at a constant centrifugal acceleration of 120 g in the centrifuge, its strain fluctuates, indicating that the slope has been damaged at this time and partial damage has occurred at the toe of the slope. Under the influence of self-weight, a small range of sliding occurs, and bending tensile cracks occur.

### 3.2. Numerical Model Test

(1)　Deformation and failure process of the slope

The failure process obtained by the Y-Mat program is shown in Figure 11. Under the condition of self-weight, the soft metamorphic rock slides along the layer, causing local subsidence at the top of the slope (Figure 11b). After excavation and rainfall, the rock layers near the slope toe undergo tension failure parallel to the bedding surface. Additionally, these layered rocks begin to bend, topple, and collapse (Figure 11c). The curved rock layers undergo local tension damages that are almost perpendicular to the bedding layer (Figure 11d). As the deformation intensifies, local tension cracks expand upward and gradually penetrate, forming a unified but uneven sliding surface (Figure 11e). Additionally, the slope eventually takes overall instability (Figure 11f).

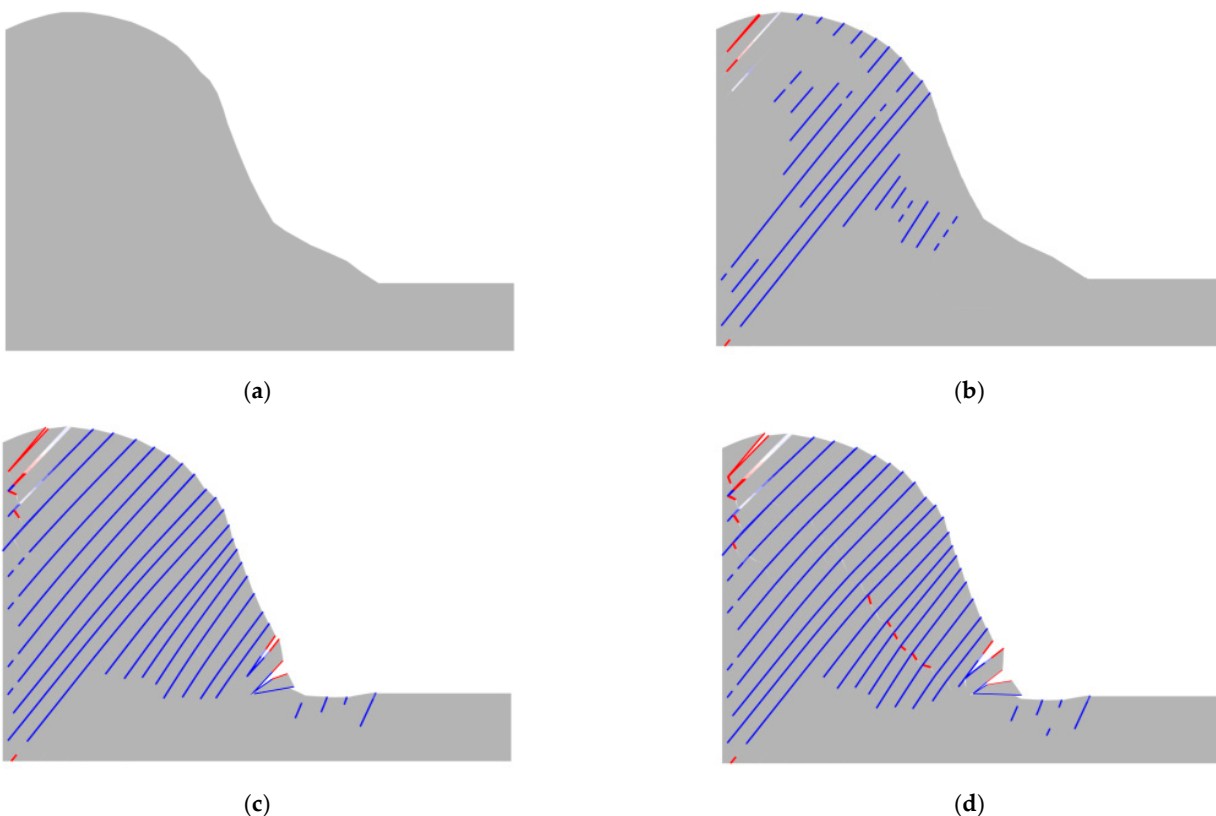

(**a**) (**b**)

(**c**) (**d**)

**Figure 11.** *Cont.*

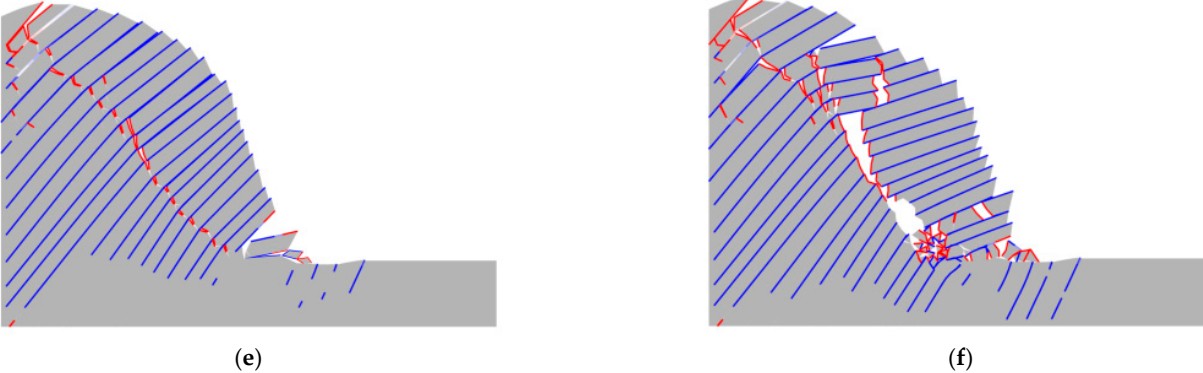

(**e**)　　　　　　　　　　　　　　　　(**f**)

**Figure 11.** Failure process of anti-dip layered soft metamorphic rock: the red line represents tensile failure, and the blue line represents shear failure; (**a**) Initial model; (**b**) Sliding between layers; (**c**) Flexural toppling; (**d**) Partial bending and breaking; (**e**) Fracture surface penetration; (**f**) Overall instability.

(2)　Monitoring results and analyses

- Stress

Figure 12 shows the distribution of the maximum principal stress, minimum principal stress, and shear stress of the slope at the partially bending and breaking stage during the simulation process (Figure 11d). The maximum principal stress nephogram shows that the internal stress of the slope decreases significantly, and even tension stress occurs locally (Figure 12a). The minimum principal stress nephogram shows that tension stress appears obviously on the surface of the bending rock layer, indicating that a tension crack may occur (Figure 12b). The three stress field distributions, including the shear stress nephogram (Figure 12c), all show significant stress differentiation and concentration at the location of failure or impending failure.

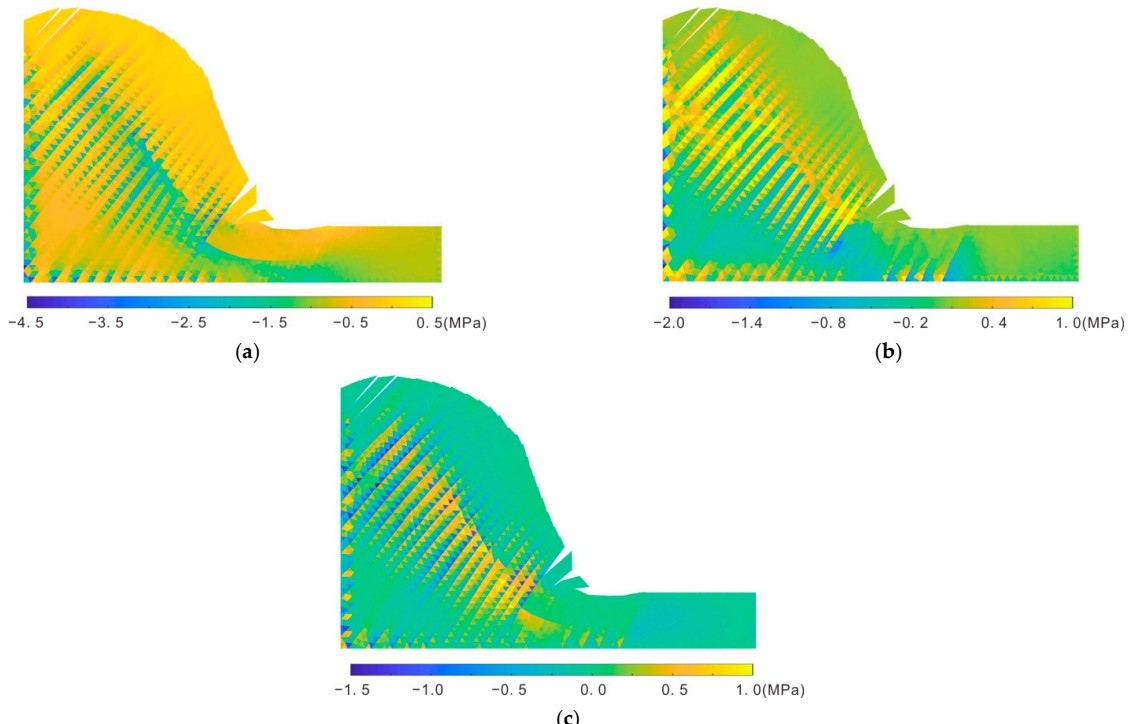

**Figure 12.** Stress distribution during partial bending and tension cracking stage: (**a**) Maximum principal stress; (**b**) Minimum principal stress; (**c**) Shear stress.

Figure 13 shows the stress duration curves of five monitoring points. In the figure, it can be seen that monitoring point 1 and point 4 exhibit oscillatory changes in stress due to their location near the potential fracture surface. However, eventually, the stress of the two points tends to zero due to slope damage. The stress curves at monitoring points 2 and point 3 are relatively standard. Monitoring point 2 is located in the rock above the sliding surface, and its stress value is smaller than that of point 3. Monitoring point 5 is located near the slope surface, and the stress value is always at a low level.

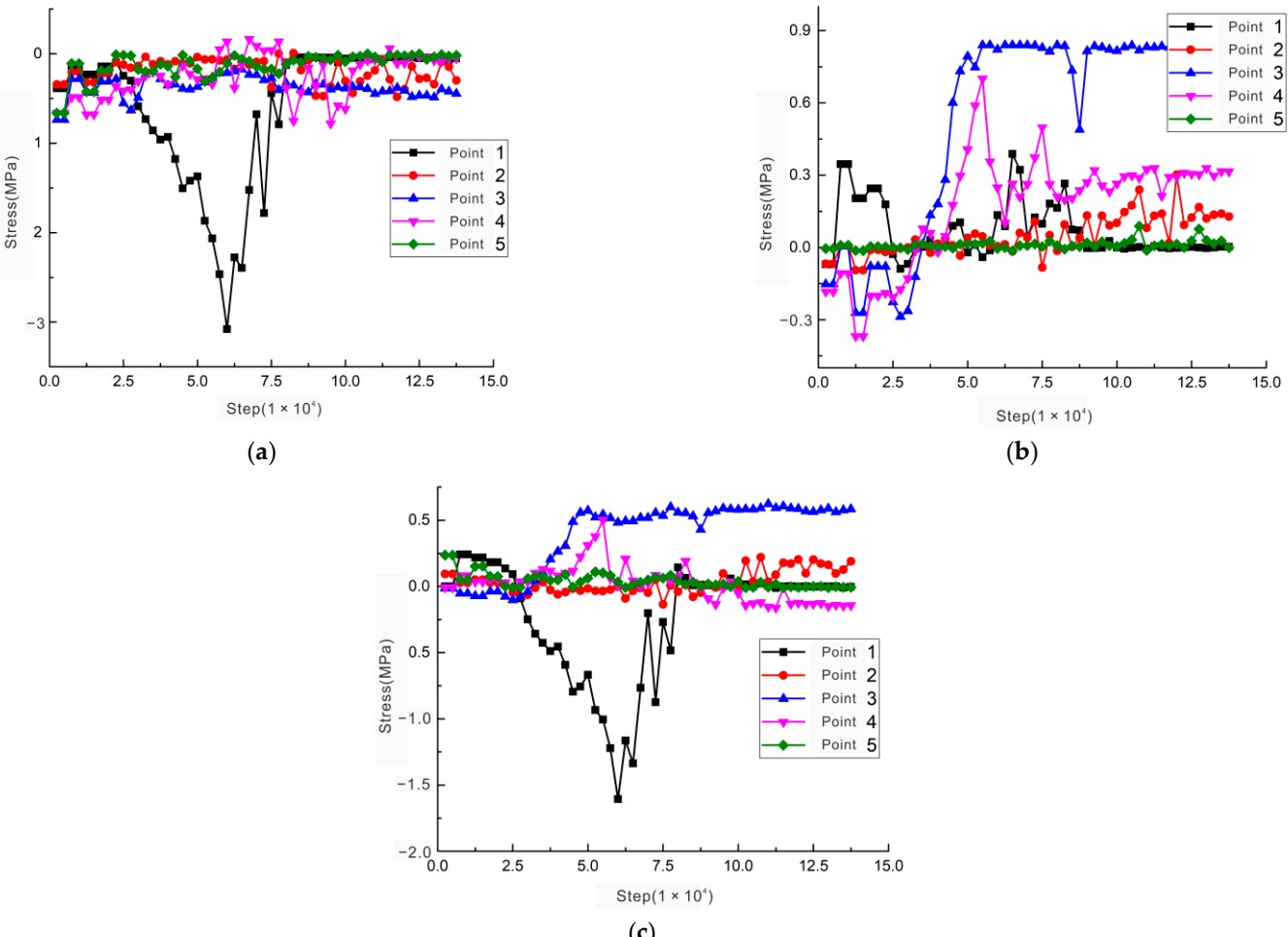

**Figure 13.** Stress duration curve of monitoring points: (**a**) Maximum principal stress; (**b**) Minimum principal stress; (**c**) Shear stress.

- Displacement

Figure 14 shows the distribution of the slope horizontally, vertically, and total displacement at the fracture surface penetration stage during the simulation process (Figure 11f). According to the figure, the maximum horizontal displacement of the anti-dip layered rock slope is 3.7 m, the maximum vertical displacement of the slope is 2.8 m, and the maximum total displacement of the slope is 4.5 m. From the perspective of displacement, the instability type of this slope is a landslide. The rock mass near the fracture surface experiences a significant displacement mutation.

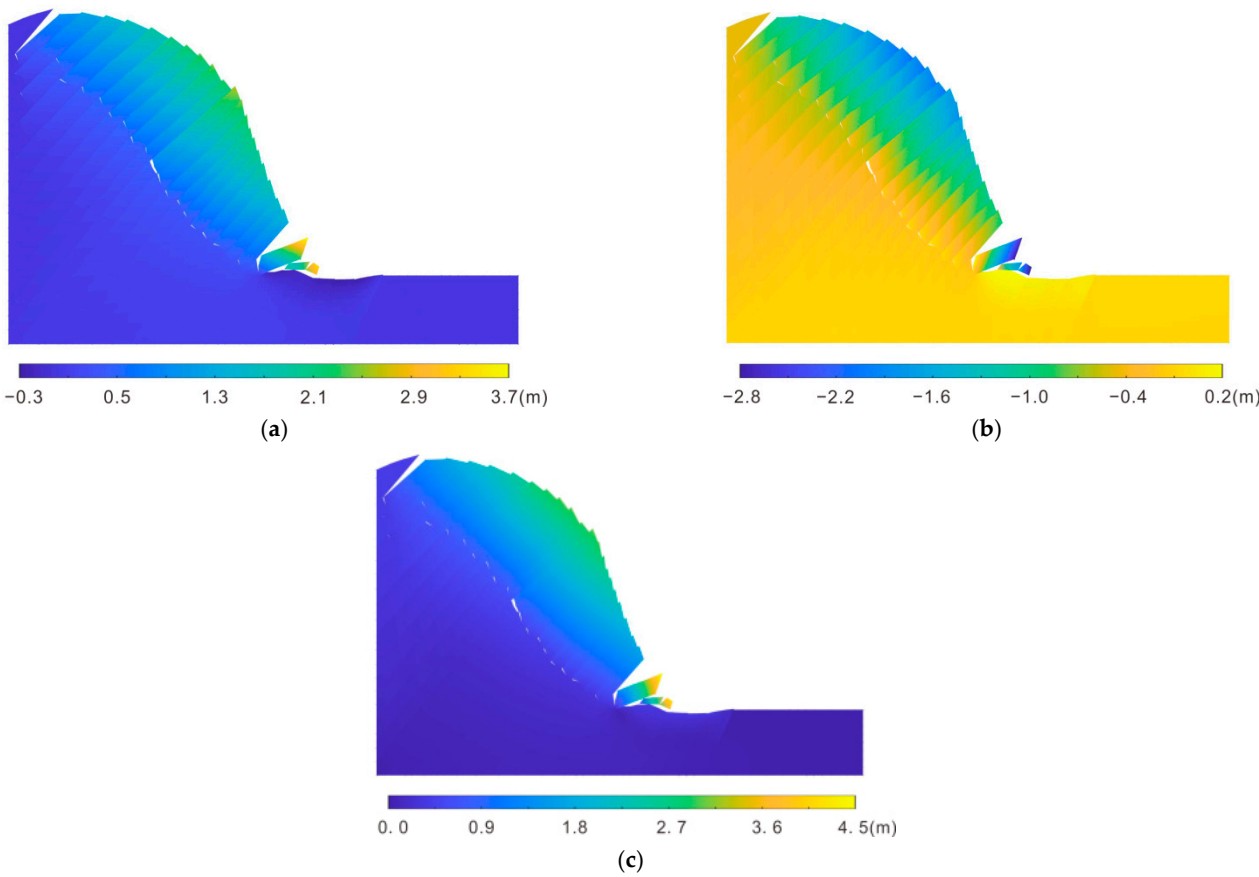

**Figure 14.** Displacement distribution during fracture surface penetration stage: (**a**) Horizontal displacement; (**b**) Vertical displacement; (**c**) Total displacement.

Figure 15 shows the displacement duration curve of five monitoring points. It can be seen in the figure that the displacement duration curves of the monitoring points have obvious regularity, with three typical stages. That is to say, the displacement of monitoring points first slowly increases, then increases at a constant rate, and finally tends to increase steadily at a small rate. Additionally, the closer the slope surface is, the faster the displacement of the slope increases and the greater the displacement value. Similarly, the closer the slope toe is, the same rule is observed for the increase in the rate and magnitude of slope displacement. Unlike landslides that slide along layers, due to the bending and toppling of rock layers, the maximum deformation and earliest failure location of the anti-dip slope usually appear near the slope surface. Additionally, due to the influence of excavation, the closer it is to the excavation area at the foot of the slope, the earlier and greater the deformation.

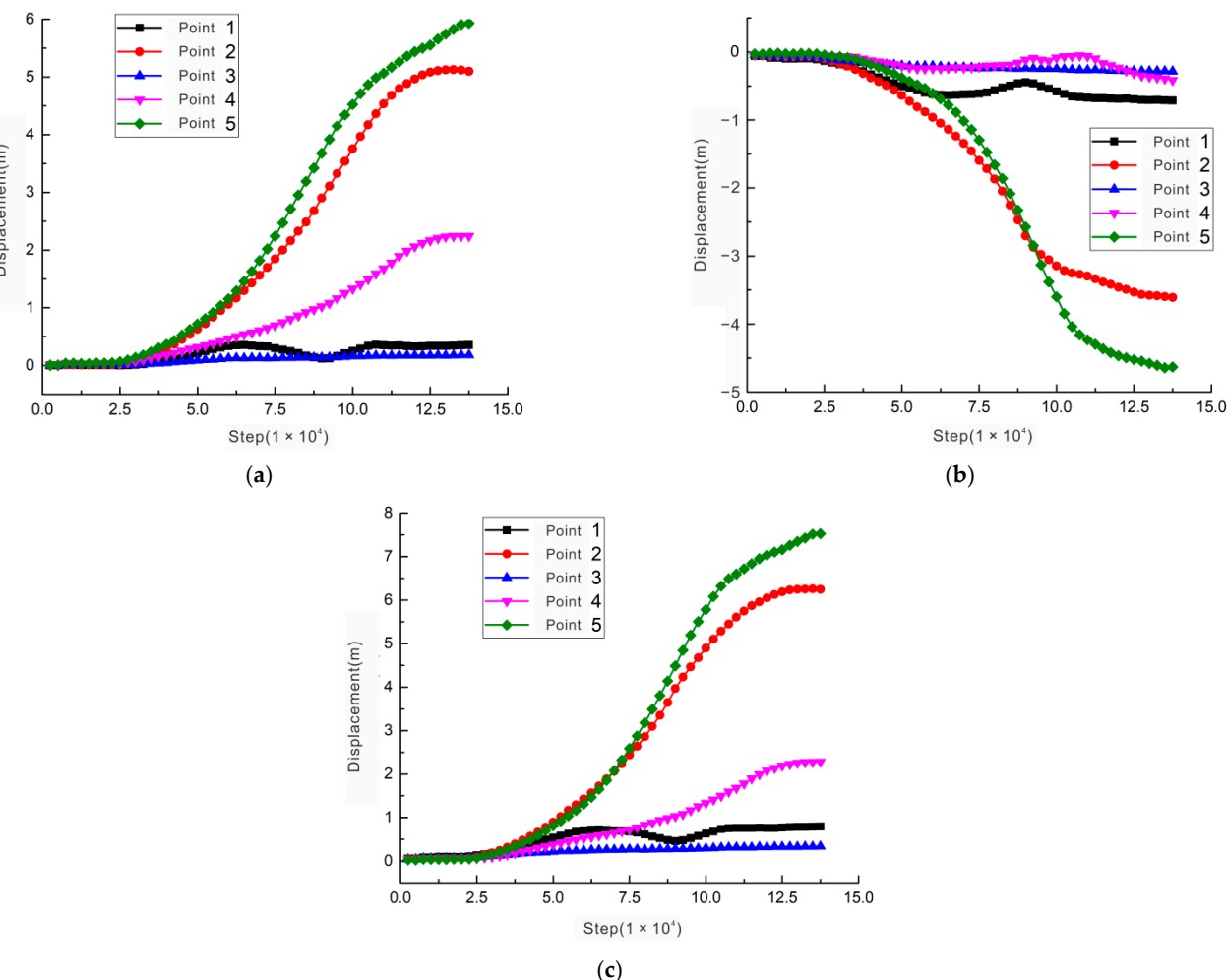

**Figure 15.** Displacement duration curve of monitoring points: (**a**) Horizontal displacement; (**b**) Vertical displacement; (**c**) Total displacement.

## 4. Failure Mode of Anti-Dip Layered Soft Rock Slope

Based on the results of model tests and numerical simulations, the evolution mode of deformation and failure of anti-dip soft rock slopes is summarized, as shown in Figure 16. The red line denotes the tensile fracture. The whole failure process can be summarized as follows: interlayer tensile cracking–bending–toppling–breaking–collapse. Under natural conditions, the slope is relatively stable. However, under the influence of rainfall, excavation at the slope toe, weathering, and other factors, the slope begins to deform and become unstable. The slope is a steep to steep inner layered slope. Under the influence of various factors, such as the gravity of the slope, a cantilever beam is bent from the front edge toward the free direction. The bent beams are staggered with each other and accompanied by tensile cracks between layers and tensile fractures almost perpendicular to the layers. The tensile fractures appear at the rear edge of the curved body and gradually develop towards the slope. The first failure occurred at the toe of the slope. Due to excavation, the degree of freeness increased, and the front edge of the slope deformed heavily and slid towards the free surface. The locking point of the failure surface is near the slope toe, where the stress is concentrated and propagates upward, creating fractures on the slope surface. As the fractures penetrate, the sliding surface gradually forms. When the inclination angle of the sliding surface in the free direction is sufficient to make the sliding force of the overlying rock mass exceed the actual shear resistance of the surface, the rock mass rotates, bends, and topples during the stage of fracture and crushing at the root of the slab beam. As soon as the trailing edge removes the failure surface, the landslide mass

quickly slides down. The whole deformation and failure processes are extremely short and sudden. The subsequent failure, based on the previous failure, exhibits a progressive regressive failure mode, resulting in overall instability of the anti-dip rock slope.

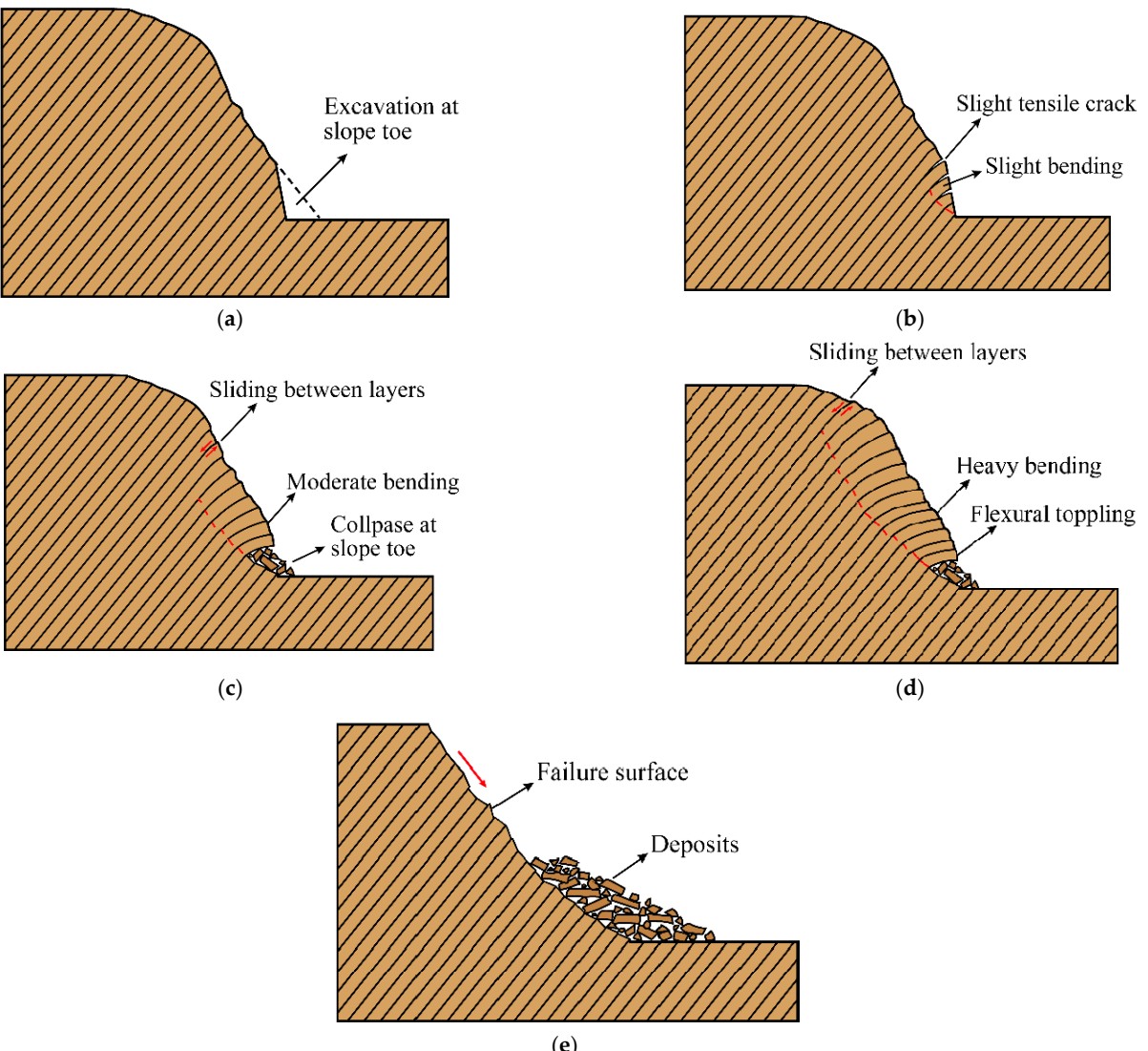

**Figure 16.** Evolution pattern of deformation and failure of anti-dip soft rock slope: (**a**–**e**) represent different stages of instability evolution.

## 5. Application and Limitation

Based on geological hazard investigation, a physical model test, and numerical simulation, this paper studies the instability process and mode of an anti-dip soft metamorphic rock slope. The obtained results of the failure process can provide scientific suggestions for the judgment of the anti-dip slope failure stage and the prediction of potential failure surfaces after highway and building slope cutting. Furthermore, it can provide a certain reference for the engineering treatment of such slopes (e.g., the selection of a support depth and location).

In the process of physical model testing and numerical simulation, an equivalent simplified method was adopted for simulating rainfall. This is somewhat different from the actual rainfall process. In the later stage, in-depth research will be conducted on the simulation of rainfall. This experiment and simulation used equal-thickness rock layers, and the actual rock layers may be unequal. This aspect of research will be considered in subsequent studies.

## 6. Conclusions

The large-scale centrifuge model test and numerical model test were used to determine the deformation and failure mechanism of anti-dip layered soft rock slopes. The main conclusions can be drawn as follows:

(1) The centrifuge physical model test can effectively simulate the process of anti-dip soft rock slopes after rainfall. At a slope of 50° and an inclination of 60°, as the acceleration of the centrifuge gradually increases (0–120 g), the slope gradually collapses from local to overall.

(2) The Y-Mat numerical simulation program based on the combined FDEM has obvious advantages in simulating the toppling, bending, tensile fracture, and sliding of an anti-dip soft rock slope. The results show good consistency with the experimental results obtained from the physical model.

(3) The deformation process and failure mode of the anti-dip soft rock slope occur under rainfall and excavation conditions. First, under the action of gravity, the rock layers bend. Second, the rock strata undergo tension failure parallel to the bedding surface. Third, the rock strata continue to bend and topple, and the rock strata undergo bending tension failures that are almost perpendicular to the rock layers. Then, the bending and cracking phenomenon gradually extends toward the upper part of the slope. Local tension failures gradually penetrate to the top or shoulder of the slope. Finally, the rock above the penetrating tension fracture surface eventually slides downward along the surface, causing slope instability.

**Author Contributions:** J.J.: conceptualization, methodology, writing—original draft. X.P.: methodology, centrifuge model data analysis. G.L.: software, numerical model data analysis, writing—review and editing. G.C.: funding acquisition, writing—review. X.G.: supervision, writing—review and editing. B.H.: supervision, writing—review and editing. All authors have read and agreed to the published version of the manuscript.

**Funding:** This study was funded by the National Key Research and Development Program of China (Grant No. 2018YFC1504702), the National Key Laboratory Open Fund Project of China (Grant No. SKLGP2015K007), and the Geological Survey Project of China Geological Survey, the Ministry of Natural Resources of China (Grant No. DD20221739).

**Institutional Review Board Statement:** Not applicable.

**Informed Consent Statement:** Not applicable.

**Data Availability Statement:** The data used to support the findings of this study are available from the corresponding author upon request.

**Conflicts of Interest:** The authors declare no conflict of interest.

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
