# Peer review of "Failure Mechanism of Anti-Dip Layered Soft Rock Slope under Rainfall and Excavation Conditions"

_sustainability, doi:10.3390/su15129398_

Round 1
Reviewer 1 Report
Based on geological phenomena observed in the field, this paper adopts physical model test and numerical simulation test to study the deformation and failure mechanism of anti-dip layered soft rock slope. The large centrifuge model test with a large loading range (250 g) and the FDEM numerical simulation test that can simulate cracking are adopted, which have innovation in methods. The results obtained from the two simulations are highly consistent and correspond well with the deformation and failure phenomena in the field. That is, results are reasonable and reliable. The deformation and failure mode of the anti-dip slop summarized by the authors could deepen the understanding of its instability mechanism and have certain guiding significance for slope prevention and control of such slopes.
The paper has reached the publishing level as a whole, but the following issues need to be noted. It is recommended to accept after minor revision.
1. Similar materials are quartz sand, gypsum and cement, and the proportion relationship is given. However, What is the basis for this ratio?
2. The Conclusion should be revised. What are the findings that are summarized by means of experiments and numerical simulations, respectively?
3. What are the elements and Bonding crack elements?
4. The advantages of the combined finite-discrete element method should be stated in the Conclusions.
5. How to consider the effect of rainfall on the strength of soft rock in the numerical modeling? Add the rock parameters in the numerical simulation after rainfall.
6. The paper consists of Methods and then Results (Centrifuge model test and Numerical model test). It is recommended that the Sections are arranged as Centrifuge model test, Test results, Numerical simulation, and Numerical results.
7. In Introduction, the advantages of the new numerical simulation method are not addressed thoroughly. Please add some references. What are the DDA, CDEM, and FDEM?
8. In Figure 7b, there are no interface between layers. In the numerical model, how to consider the interface of interlayer. In Figures 13 and 15, the position of monitoring points should be presented in the figure, as shown in Figure 7. The letters in Table 2 should be capitalized, such as length, which should be written as Length. Table 4 needs to be modified to a three-line table.
9. Authors'institution. Secondary institution should be placed first, such as State Key Laboratory of Geo-hazard Prevention and Geoenvironment Protection, Chengdu University of Technology, Chengdu, 610059, China.
10. The Figure format needs to be unified. For example, if a figure contains two or more sub figures (a, b, c, etc.), the calibration positions of the sub image names are inconsistent. Figure 4 is labeled at the top left, Figure 7 is labeled at the bottom, and Figure 3 is not marked. It is recommended to modify and unify according to the format specified by the journal.
11. It is recommended to enlarge the figures and decrease the font size, such as two photos in Figure 2, the font inside the image in Figure 8.
12. A space should be added between numbers and units, such as the description of centrifuge acceleration: 120 g.
It is recommended that the author carefully check the spelling of the following words, grammar problems
Author Response
Dear reviewer,
The authors have responded one by one according to your feedback and made modifications in the corresponding positions. Specific content can be seen in the attachment.
All authors

Reviewer 2 Report
1- This manuscript has good innovation, but the author did not announce it well at the end of the introduction. The objectives are not clear and the literature review is also a little weak. It is better for the author to use more recent references.
2- Figure 15 discusses the reason for the change in monitoring points.
3-Also, other sections are also poorly discussed
4- In conclusion: The authors have mixed several issues which are not within the scope of their present study. Neither the study limitations nor recommendations are provided in this study. This section should be re-written. The authors should be guided by components of a good discussion and conclusion remarks.
ï‚· Interpretation of the results: What do the results imply in the context of your study;
ï‚· Implication: why do the results matter to the problem being addressed;
ï‚· Limitations: what can’t the results tell us; based on the methods, data etc
ï‚· Recommendation: what practical actions or scientific studies should follow?
Moderate editing of English language
Author Response
Dear reviewer,
The authors have responded one by one according to your feedback and made modifications in the corresponding positions. Specific content can be found in the attachment
All authors

Reviewer 3 Report
1. The abstract is required to be supported by Some quantitative results.
2. The manuscript needs proofreading. The presentation style and sentence structure shall be the same throughout the manuscript.
3. The research objective statement is not clear. The research objectives must be clearly elaborated.
4. How the research objective targets the research gaps in the field of discussion.
5. The mineralogy of the shist under consideration shall be tabulated. Is XRD data available?
6. Is there a layer of superficial soil in real life if so, how it affects the slope stability, and why it has been ignored in the model?
7. Cementation effects play an important role in rock slope stability. Usually, it’s very difficult to consider it in reconstituted models. How this effect has been incorporated.
8. Crack propagation increases water infiltration (in the case of the Humid zone as in the considered case), ultimately reducing the friction between the layers, and triggering the failure. The variation in moisture vs friction (shear resistance) cannot be ignored as such. The author’s opinion in this regard is of great importance.
9. Further study opportunities and field applications are missing. What are the recommendations for the construction industry (i.e., Highways and retaining walls)?
The manuscript needs proofreading. The presentation style and sentence structure shall be the same throughout the manuscript.
Author Response

(The authors gave the same response as above.)

Reviewer 4 Report
The author has made a good attempt to understand the "Failure mechanism of anti-dip layered soft rock slope under 2 rainfall and excavation conditions". However, a few comments are mentioned below:
1. Existing kinds of literature are missing
2. Scope and objective are defined effectively
3. Sample collection details are missing
4. methodology reference is missing
5. Table 2. Scale factors used in centrifuge model tests reference is missing how you calculated these parameters, is it secondary or primary? provide the details
6. How you decided on these dimensions Figure 3. Model construction: (a) top view of slope model; (b) completed model in the test box
7. Figure 9 results discussion is missing
8. In Figure 10, why acceleration values are not matching with existing data
9. Factor of safety values is missing.
10. Conclusion should be very precise
11. Recommondation and signifance is missing
Major
Author Response

(The authors gave the same response as above.)

Round 2
Reviewer 4 Report
OK
OK